# A qualitative study of health visitors' family focused practice with mothers with mental illness in Northern Ireland: Perspectives of health visitors, mothers and partners

**Anne Grant** ⓘ *, **Rachel Leonard, Mark Linden** ⓘ

School of Nursing and Midwifery, Medical Biology Centre, Queen's University Belfast, Belfast, Northern Ireland, United Kingdom

* a.grant@qub.ac.uk

## Abstract

**Data Availability Statement:** All relevant data are within the paper.

**Funding:** The author(s) received no specific funding for this work.

### Background

Despite benefits of family focused practice, little is known about health visitor's practice with families when mothers are mentally unwell. Health visitors are midwives and nurses with additional training in community public health.

### Objectives

To explore multiple perspectives of health visitor's family focused practice with families when mothers have mental illness in Northern Ireland.

### Methods

Ten health visitors, 11 mothers with mental illness and seven partners completed in-depth interviews in Five Health and Social Care Trusts. Participants were asked to describe their experiences of providing or receiving family focused practice within health visiting and data was analysed using thematic analysis.

### Results

Health visitors primarily addressed mothers and children's needs rather than also supporting partners. Additionally, they only addressed mother's needs associated with less severe mental illness (i.e. postnatal depression). Health visitors and mothers converged on many issues, including the influence of the health visitor's personal and professional experiences on their practice, central role of the relationship between health visitors and mothers and importance of health visitors supporting partner's well-being. While partners did not perceive that health visitors should support their well-being they expressed a need for further information and knowledge in order to support mothers.

**Competing interests:** The authors have declared that no competing interests exist.

## Conclusion

Health visitor's practice largely centres around mother and baby. For health visitors to increase their family focused practice they need to meet needs of mothers who have serious mental illness more effectively and consider how partners can be included in their practice, in a manner that is beneficial and acceptable to them. This study contributes to better understanding of health visitor's family focused practice with mentally ill mothers and highlights the need for more effective engagement with mothers with serious mental illness and partners. It also highlights that for health visitors to engage in family focused practice they need the necessary training and time to do so. Results can inform organisational developments in family focused practice within health visiting.

## Introduction

A significant percentage of women who are pregnant (10%) and have children (13%) suffer mental illness; most notably anxiety or depression [1]. There is a higher level of maternal mental illness in Northern Ireland (NI) than elsewhere in the United Kingdom (UK) [1, 2]. Abel et al. (2019) reported that 53% of children aged over 16 had mothers who experienced common mental illness (i.e. neurotic [minor] disorders such as depression or anxiety) or serious mental illness (SMI) (i.e. schizophrenia).

Multiple international studies highlight the adverse impact that maternal mental illness can have for mothers who have mental illness and their families, with a strong correlation between a mother's mental illness and her children's wellbeing [1–3]; all aspects of family life and children's development and wellbeing can be negatively impacted [2, 4, 5]. Equally, stress of motherhood can exacerbate mother's mental health [6, 7]. Other adult family members, including partners may also be negatively affected [8, 9]. Consequently, health professionals are encouraged and in some cases mandated to engage in early intervention strategies with mothers who have mental illness and their families [5].

Health visitors are well placed to play a key role in supporting these families [9] if they can engage in Family Focused Practice (FFP) [10]. Roughly, 10,800 nurses and midwives in the UK have additional specialist community public health training and practice as health visitors [11]. This specialist training and experience is centred on health promotion, perinatal and infant mental health, child health and education [8, 12]. Health visitors can refer directly to general practitioners and deliver a universal service for families with children from birth to aged four [8, 12]. Their core programme is centred around child health promotion, and family well-being, though they also give further support to families who have additional needs, including when mothers have mental illness [13].

Family focused practice accommodates and supports the inter-related needs of family members [6]. In applying this approach to practice in the context of health visiting it would entail health visitors supporting not only mothers and their young children (i.e. 0–4) but also older children and partners and other adult family members [9, 10]. Health professionals' FFP may be broad in scope and extent, ranging from, at a lower level, identifying service users' families' needs and referring them to other services, to a higher level, supporting them directly. For instance, higher levels of FFP entail health professionals actively supporting family members by engaging them about their concerns related to parental mental illness and directly intervening with them to promote their capacity to cope, using strategies such as psycho

education [6]. Psycho education entails providing written, verbal, and/or online information to facilitate mothers, partners and children to understand mental illness and how to cope with it [14].

Despite increasing evidence regarding benefits of FFP [6, 15, 16] and recommendations for health visitors to engage in it [13, 17], knowledge is scarce about health visitors' practice with families affected by maternal mental illness [18]. This study adds to existing literature as it represents a first attempt to explore multiple perspectives (i.e. health visitors, mothers, partners) of FFP within health visiting in the context of service delivery for mothers who have mental illness. This study conducted in NI aimed to explore health visitors' perceptions regarding their ability to meet needs of mentally unwell mothers and their families, including partners. It also aimed to explore the acceptability and adequacy of this support from health visitors, mothers and partners' perspectives.

## Design and methods

Qualitative findings from a larger mixed methods study, which examined health visitors' FFP with mentally ill mothers in NI [8] are reported in this paper.

The first phase entailed a cross sectional survey of health visitors (N = 488) practicing across NI. The Family Focused Mental Health Practice Questionnaire (FFMHPQ) employs a seven-point Likert Scale (ranging from strongly disagree to strongly agree) to measure individual professional and organisational factors associated with FFP (i.e. family focused training) [19, 20]. The initial measure, was developed by Maybery et al. [20] in Australia for use with a variety of professional disciplines. It consists of 16 subscales and 45 items and has good content and construct validity and generally good internal subscale reliability. Psychometric information on the original FFMHPQ subscales is detailed elsewhere [20]. As the FFMHPQ had never before been used in a population of health visitors, the second author undertook an exploratory factor analysis (EFA) which created a more parsimonious instrument consisting of 20 items, across two factors named as professional and organisational influences on FFP. The internal consistency of this newly proposed 20-item version, was found to be excellent with a Cronbach's alpha of 0.949. Please see Leonard et al. [21] for further details of the EFA and Leonard et al. [22] for the study protocol. The recruitment of health visitors occurred between 05/09/2017 and 07/10/2018.

The second phase employed a qualitative descriptive design. Individual semi-structured interviews were undertaken with 10 health visitors who obtained the highest and lowest FFP scores on the FFMHPQ to further explore and expand upon findings (lowest score = 59, highest score = 140). Mothers with mental illness (*n* = 11), who received health visiting services, and their partners (*n* = 7) were also interviewed. The interview findings are reported here with regard to health visitors, mothers and partner's experiences of FFP.

### Ethical considerations

A National Health Service Research Ethics Committee granted approval (Ref 17/WS/0131) in June 2017. Participants received written and oral information about the study and completion of a written informed consent form was required by all participants prior to the interview. Maintaining participants' confidentiality is often a major ethical concern of interpretive research because of the intimate nature of the research [23], but was maintained through the use of pseudonyms and changing specific contextual details that could possibly reveal the identity of the participant. The study was conducted in accordance with the statement of ethical practice and standards set out by the Declaration of Helsinki and in line with current

university processes and regulations to ensure ethical considerations in relation to integrity, the voluntary participation of participants and confidentiality were followed throughout.

## Sampling procedure and participants

Health visitors were purposively selected in health visiting services across NI to participate in interviews. Twelve of 35 health visitors who agreed to contribute were invited to participate in accordance with their score on the FFMHPQ [20]. Health visitors with the highest and lowest (lowest score = 59, highest score = 140) scores were targeted to facilitate inclusion of those who reported undertaking most and least FFP [21] and ten were then recruited and interviewed. Health visitors completed an interview volunteer form to consent to undertaking interviews [21]. Those with current caseloads, at least six months post registration experience and permanent positions met inclusion criteria.

Purposeful sampling was also used in the five health care trusts, to select two mothers receiving services from a health visitor and two partners of women with mental illness in receipt of this service. Mothers fulfilled inclusion criteria if they received health visiting services; had mental illness before they were pregnant, or during and after (puerperium period) pregnancy and were able to provide informed consent. Inclusion regarding the mother's mental health was assessed through examination of medical records by their health visitors. Partners were required to be aged over 18 years and have adequate understanding of the English language and did not require to be in a relationship with those mothers who participated in interviews [21].

Initial sample sizes between 10–12 participants per group were based on guidelines from the literature [24] and dependant on data saturation. A key contact was assigned in each Trust (a local collaborator at managerial level) who assisted the second author to circulate information packs to mothers and partners via two health visitors. Information included detail regarding eligibility criteria and recruitment instructions. Subsequently, health visitors invited mothers and partners who met inclusion criteria. Those who wished to take part sent the interview volunteer form back to the second author by post. No one in the research team had previously worked with participants. None of the participants invited declined or dropped out. Saturation was met for all three samples.

## Data collection tool and methods

A qualitative approach was deemed necessary due to the limited research exploring health visitors, mothers and partners' perspectives of FFP. Consequently, a semi-structured interview schedule was developed for each group of participants. These drew upon findings of a qualitative systematic review that explored FFP in health visiting in the context of maternal mental illness [21], along with findings from the survey.

Questions in the three interview schedules focused on (1) families' needs and health visitors' response to meeting those needs and (2) health visitors' ability to support family members. The interviews explored what activities health visitors undertook with families, the challenges they faced and experiences in being family focused. The questions were tailored accordingly (i.e. for the health visitor, mother or partner). For additional detail regarding topic guides see Table 1 for health visitors, Table 2 for mothers and Table 3 for partners.

Single, one to one semi-structured interviews were undertaken by the second author (female with experience of qualitative interviewing) with all participants between May and September 2018 and generally lasted 60 minutes. Interview duration ranged from 31–62 minutes for mothers, 30–46 minutes for partners and 37–70 minutes for health visitors. Health visitors' interviews were undertaken in their place of employment and those with mothers and partners took place in their homes. No one else was present besides the participants and

**Table 1. Health's visitor topic guide.**

| | |
|---|---|
| Demographics—I first would like to know: | Length of time qualified as a health visitor?<br>During CPD or other training have you specialised in anything? (ie. CBT, safeguarding)<br>Have you had any (1) family focused education/training, (2) domestic violence training, and (3) think family training (4) child focused training? |
| Identification of mental illness | Can you describe the process of how you identify a mother's mental ill health? Assessments, multi-disciplinary information sharing?<br>What do you think the needs are of a mother with mental illness/ her children/ partner?<br>Her relationship with her children and spouse?<br>What do you perceive your role to be in meeting the needs you have identified? |
| Family focused activities: | What is your understanding of the term family focused practice, and how does it relate to your role, if at all?<br>Do you think that a mother's mental illness impacts on her ability to parent? If so, how?<br>How do you think being a parent could impact on a mothers' MI?<br>Do you have contact with a partner, when a mother has a mental illness? Can you describe this? What does this entail?<br>How much contact do you have with the children, when a mother has a mental illness? Can you describe this? What does this entail?<br>How confident are you at discussing the impact of the mothers' mental illness on her family with (1) the mother, (2) her partner (or other family), (3) her children. Can you describe this?<br>How much contact do you have with other services that the family are involved with, if any? Do you discuss the mothers mental illness? |
| Capacity to undertake family focused practice? | Do you think that the care you provide meets the needs of mothers who have mental illness? The needs of their children, partner and families? If yes, please explain further?<br>If not, why are these needs not being met? How, in your opinion, could the needs of these mothers be met? Their children, partner and families?<br>How is your assessment and response different, if at all, dependant on whether the mother has a pre-existing mental illness or has developed mental illness postnatally?<br>Do you think that you have the capacity to meet the needs of mothers with any mental illness? If you feel more confident in meeting the needs of certain mental illness then why?<br>What factors facilitate health visitors in meeting your needs and that of your children and partner? Can you tell me more about that?<br>What are the current challenges, if any, that you face when you are working with a mother who has mental illness or substance misuse? Their child(ren)? Partner or family?<br>How might these challenges be improved?<br>How might health visiting services for mothers, their children and families be further developed, if at all? |
| Closing questions | Are there any topics which I did not address which you would have liked, or expected, me to have asked/discussed? |

researcher. Participants received a payment in thanks (£15 gift voucher) to thank them for their time and any inconvenience caused by being involved.

A digital audio recorder was used to record interviews which were then written verbatim. All electronic information was stored on a password-protected computer. Participants were asked to review their transcript to remove identifying information, to include relevant information and/or amend inaccuracies (member checks; [23]). No participants chose to revise their responses and/or to provide additional information.

## Data analysis

Data was analysed using thematic analysis, which allowed for a flexible approach to identifying, analysing and reporting patterns in the data [25]. To assist in analysing the multiple

**Table 2. Mothers' topic guide.**

| | |
|---|---|
| Demographics—I first would like to ask you: | Your age, and how many children and their ages?<br>Could you describe to me briefly your current and past mental health problems? (type, duration and severity)<br>Are you currently (or have in the past) receiving any other services beside health visiting (eg. Mental health services, family support)? If yes, can you explain why those services are involved?<br>Can you tell me briefly what your main support networks are? (i.e. Parents, partner, friends, neighbours) |
| Needs—of the mother and/or their families: | From the issues you have identified, can you tell me what your needs are in relation to mental health (domestic violence/substance abuse, if it comes up)?<br>What impact, if any, do you think your mental health has on your parenting?<br>What impact, if any, do you think being a parent has on your mental health?<br>What impact, if any, do you think your mental health problems have on family functioning? (i.e. roles within the house, house hold task, childcare)<br>What you believe your needs are as a mother with mental health issues?<br>What needs does your partner/ your children have in relation to your mental illness? |
| Health visitors' capacity to engage in FFP: | Based on the needs you have identified do you think that health visitors meet (a) your needs as a mother who has mental health issues (and other issues that have been identified), (b) the needs of your children, (c) the needs of your partner, and (d) the needs of other family members? Can you tell me more about this?<br>If your family is involved with other services, how often would your health visitor link in with them? Do they discuss your mental illness?<br>How do you think health visitors could best meet your needs, and that of your children and partner?<br>What factors facilitate and/or hinder health visitors in meeting your needs and that of your children and partner? Can you tell me more about that? |
| How health visitors' capacity to engage in FFP may be further developed? | What might help health visitors in working with partners, children and mothers with mental illness? Give examples.<br>On the basis of your experience what skills and behaviours do health visitors need to be able to effectively support you and your family?<br>What knowledge do health visitors need to be able to effectively support you and your family? |
| Closing questions | Are there any topics which I did not address which you would have liked, or expected, me to have asked/discussed?<br>Is there anything else you would like to ask me? |

perspectives data were analysed on a group level [26], with interviewees treated as distinctive groups of individuals (e.g. Health visitors, mothers, and partners) [21]. NVivo qualitative data analysis software version 10.0 [27] ensured organisation of the data and methodological rigour [21]. In reporting findings, each participant was identified by a number, their health service trust (i.e. N = Northern, S = Southern, SE = South Eastern, W = Western, B = Belfast) and whether they are a health visitor (HV), mother (M) or partner (P) [21]. The age of mothers and partners and number of children are also included. This detail illustrates that quotes were used from a range of participants across the three groups. Interviews with all three groups of participants were conducted until data was saturated. We define saturation as the point by which no additional data are being found to develop further themes [28].

**Table 3. Partners' topic guide.**

| | |
|---|---|
| Demographics—I first would like to ask you: | Your age, and how many children and their ages?<br>Could you describe to me briefly your partner's current and past mental health problems? (type, duration and severity, service use [AMHS/FIT])<br>Can you tell me briefly what your main support networks are? (i.e. Parents, partner, friends, neighbours) |
| Needs–you, your partner and children: | What impact, if any, do you think your partner's mental health has on their parenting?<br>What impact, if any, do you think being a parent has on her mental health?<br>What impact, if any, do you think your partner's mental health has on your own mental health? (Well-being, parenting, life in general)<br>What do you think your needs are in relation to living with a partner who has a mental illness? What do you think your partners' needs are? Your children's?<br>What impact, if any, has your partners mental illness has on family functioning? (ie. roles within the house, house hold task, childcare)<br>Do you think you have adequate knowledge on your partner's mental illness? If no, what information would you like? |
| Health visitors' capacity to engage in FFP: | Based on the needs you have identified do you think that health visitors meet (a) your needs as a partner, (b) the needs of your children, and (c) the needs of your partner (d) other family members? Can you tell me more about this?<br>If your family is involved with other services, how often would your health visitor link in with them? Do they discuss your partner's mental illness?<br>How do you think health visitors could best meet your needs, and that of your children and partner?<br>What factors facilitate and/or hinder health visitors in meeting your needs and that of your children and partner? Can you tell me more about that? |
| How health visitors' capacity to engage in FFP may be further developed? | What might help health visitors in working with partners, children and mothers with mental illness? Give examples.<br>On the basis of your experience what skills and behaviours do health visitors need to be able to effectively support you and your family?<br>What knowledge do health visitors need to be able to effectively support you and your family? |
| Closing questions | Are there any topics which I did not address which you would have liked, or expected, me to have asked/discussed?<br>Is there anything else you would like to ask me? |

## Rigour

To ensure the rigour of the study the consolidated criteria for reporting qualitative research (COREQ) [29] (32 item checklist for interviews) were followed and qualitative research trust-worthiness criteria applied [30]. To promote credibility, the analysis was triangulated by the first and third author. The second author coded all transcripts and then the first and third author analysed all codes to ensure they were meaningful and coherent and checked the original transcripts as required. The second author also considered whether there were features of practice described by only one participant [31] and in successive interviews explored this and discussed with the third author in order to develop and refine themes [21]. Interpretation was assisted by the use of field notes written after each interview. To ensure confirmability, participants quotes are used to support interpretation of the results. To further ensure credibility and confirmability during data collection, the researcher maintained a reflective diary and considered whether her assumptions about mentally ill mothers may affect data analysis [32]. The

research protocol was peer reviewed by two academics outside of the research team and published in a peer reviewed journal [21].

## Findings

### Research participants

Health visitors were all female, ranging between age 37 to 52 years. Duration in the profession ranged from 2 and 37 years. Three had expertise in breast feeding and infant mental health. Four participants practice within rural settings and the rest within either urban (n = 3) or a mixture of both (n = 3). The majority had full time positions (n = 7) with the rest part-time (n = 3). All but one were parents (n = 9) and just over half had personal experience of mental illness (n = 6).

The 11 mothers were aged between 25 and 38 years, ten had a partner, with one being a single parent. Five mothers had one child, while five had 2 children, and one mother had three. Mental health problems ranged from PND, eating and anxiety disorders and obsessive compulsive disorder. Six of the mothers had more than one mental health problem e.g. PND plus anxiety. Five of the mothers' mental health problems started after pregnancy and had no previous history, while six had mental health problems that pre-existed pregnancy.

In total seven partners were interviewed. Partners were between 27 and 38 years of age and were all male. Four had one child, one had two children, while the remainder had three children. The partners of the fathers had a range of mental illnesses, such as; PND, obsessive compulsive disorder, eating disorders and anxiety. For additional detail of all participants see Table 4.

### Themes

Health visitors, mothers and partners described the perceived needs of families experiencing maternal mental illness and the health visitor's response and capacity to meet those needs as illustrated in Table 5.

### Theme 1: Families' needs relating to maternal mental illness

All participants discussed adverse impacts of mental illness for mothers, children and partners and associated needs. Both mothers and partners also discussed the lack of information and support for partners. The first subtheme focuses upon mother's needs in relation to bonding and stigma associated with mental illness. The second subtheme focuses upon partner's needs in relation to understanding maternal mental illness and being able to more effectively support mothers.

**Needs of mothers in relation to their mental illness and parenting.**   The mothers described various ways in which their mental health affected their parenting and the feelings this evoked. Many mothers felt that their mental health had impacted their capacity to bond with their baby and with older children. In relation to an older child, another mother stated, "You don't want to do anything with the kids. I remember [1st child] asking to play and I just wouldn't, couldn't. Because I just wanted to lie down and. . .feel sorry for myself" (SEM7, mother of two aged 26). These difficulties in bonding with their children caused intense feelings of guilt for the mothers: "The guilt of not feeling attached to (child) for the first four months is just killing me" (SEM3, mother of one aged 37). Health visitors also noted mother's difficulties in bonding and the need to help them build a secure attachment with their baby, "she em thinks the baby isn't hers. Not quite a psychosis. And she has no feeling for the baby at all" (SEHV7).

Mothers also described feeling ashamed of being mentally ill and the associated stigma which led to them not accepting that they were unwell and also trying to cover up their

**Table 4. Mothers, partners and health visitor's demographics.**

| Mothers ID | Age (years) | Partner | No. of children/ age | Mental Illness | Postnatally, antenatally or pre-existing |
|---|---|---|---|---|---|
| WM1 | 34 | Yes | 1/20 months | Postnatal Depression | After pregnancy |
| WM2 | 25 | Yes | 1/6 months | Postnatal Depression and anxiety | Pre-existing, worsened after pregnancy |
| SEM3 | 37 | Yes | 1/9 months | Postnatal Depression | After pregnancy |
| SM4 | 30 | Yes | 1/20 months | Obsessive compulsive disorder, Postnatal Depression and anxiety | After pregnancy |
| SM5 | 35 | Yes | 2/ 5 months and 3 years | Postnatal Depression and eating disorder | Pre-existing, worsened after pregnancy |
| NM6 | 38 | Yes | 3/ 8, 6 and 4 years | Postnatal Depression | After pregnancy |
| SEM7 | 26 | Yes | 2/ 6 and 1 years | Depression and Postnatal depression | Pre-existing, worsened after pregnancy |
| NM8 | 29 | Yes | 2/ 6 and 1 years | Anxiety | After pregnancy |
| BM9 | 25 | Yes | 1/ 9 months | Depression | Pre-existing |
| BM10 | 32 | Yes | 2/ 1 and 2 years | Depression, anxiety and postnatal depression | Pre-existing, worsened after pregnancy |
| BM11 | 34 | No | 2/ 5 and 2 years | Depression and postnatal depression | Pre-existing, worsened after pregnancy |

| Partners ID | Age (years) | Gender | No. of children/ age | Partners' mental Illness | |
|---|---|---|---|---|---|
| WP1 | 27 | Male | 1/6 months | Postnatal depression and anxiety | |
| WP2 | 39 | Male | 1/20 months | Postnatal depression | |
| SEP3 | 31 | Male | 1/9 months | Postnatal depression | |
| SP4 | 32 | Male | 1/20 months | Obsessive compulsive disorder, Postnatal depression and anxiety | |
| SP5 | 39 | Male | 2/ 5 months and 3 years | Postnatal depression and eating disorder | |
| NP6 | 39 | Male | 3/ 8, 6 and 4 years | Postnatal depression | |
| SEP7 | 27 | Male | 3/ 6, 3 and 1 years | Depression | |

| Health visitor ID | Age (years) | Gender | Specialism | Health & Social Care Trust | Rural/Urban location of service | Full-time /Part-time employment | Parent | Experience of Mental illness | FFP score (20–140) |
|---|---|---|---|---|---|---|---|---|---|
| NHV1 | 52 | Female | No | NHSCT | Urban and rural | Full time | Yes | Missing | 75 |
| SHV3 | 42 | Female | No | SHSCT | Urban | Part time | Yes | No | 81 |
| SEHV4 | 46 | Female | No | SEHSCT | Urban and rural | Full time | Yes | Yes | 81 |
| NHV8 | 37 | Female | No | NHSCT | Rural | Full time | No | No | 89 |
| SEHV2 | 45 | Female | No | SEHSCT | Rural | Part time | Yes | Yes | 90 |
| SEHV7 | 52 | Female | Infant Mental Health | SEHSCT | Urban | Full time | Yes | Yes | 98 |
| NHV5 | 51 | Female | No | NHSCT | Urban and rural | Full time | Yes | Yes | 111 |
| NHV10 | 48 | Female | Breastfeeding | NHSCT | Rural | Part time | Yes | No | 112 |
| SHV6 | 45 | Female | No | SHSCT | Rural | Full time | Yes | Yes | 115 |
| BHV9 | 44 | Female | Infant Mental Health | BHSCT | Urban | Full time | Yes | Yes | 140 |

deteriorating mood. As one mother described "I just didn't want to accept this is what was happening to me, cause I thought this will just consolidate all of my thinking that I am a useless mummy" (NM6, mother of three aged 38), and another indicated:

> I always had the dinner on the table for him (her husband) coming home. We went swimming, baby sensory, and (child) was happy, content you know what I mean. Dressed to kill for. I wanted to keep it as normal as possible (SM4, mother of one aged 30).

**Table 5. Themes and sub-themes.**

| Theme | Sub-theme |
|---|---|
| Families' needs relating to maternal mental illness | Needs of mothers in relation to their mental illness and parenting |
| | Partner's needs in relation to maternal mental illness |
| Health visitors' response to meeting families' needs | Health visitor's response to mothers and their children |
| | Health visitors' response to partners |
| Health visitors' capacity to engage in FFP | Health visitors' capacity to address mother's needs related to their mental illness and to support their partners |
| | Health visitors draw upon professional and personal experience to support mothers |

Finally accepting their mental illness, caused most mothers to feel very vulnerable, "there is something about exposing your real vulnerabilities, like I have never felt as vulnerable in my entire life", (NM6, mother of three aged 38). Relatedly, health visitors discussed how mothers were reluctant to take medication because of the perceived stigma. Health visitors were concerned about this as they perceived medication was necessary to improve mother's mental health, "they [mothers] have this thing in their head that they, it's an awful travesty to have medication, anti-depressants are a terrible thing you shouldn't be on them", (NHV1). Health visitors also reported that mothers tried to hide their mental illness due to shame and guilt and that this prohibited early intervention and could lead to crisis, "they're [mothers] are not willing to admit that there is something going on. And they are saying they are fine. . .until something happens", (SEHV4).

**Partner's needs in relation to maternal mental illness.** Mothers also discussed their partners needs for support as often they were required to take on extra responsibilities within the home in addition to having full-time jobs, "he would have come home and would have to wash and sterilise and make up six bottles after a full day of work. Maybe make dinner. . ." (NM6, mother of three aged 38). They also expressed concerns about the difficulties their partners have in being open about their own mental health, ". . .I see him coping, but sometimes underneath I think is he really coping. He might just put on a brave face and smile away" (SM5, mother of two aged 35).

Conversely, partners did not indicate that they needed support with their mental health but described difficulties in understanding and accepting the mother's mental illness. Many partners described feeling "something wasn't right" (WP2, father of one aged 39), but could not pinpoint what the cause was. They also reported finding it difficult to understand something that they had never personally experienced. As one partner stated:

*Understanding depression, I think is quite hard. For someone who is not depressed. Like I hear what (partner) is saying and sometimes she says things and I understand it and then there are other times where I just don't understand why you feel like that (SEP7, father of three aged 27).*

These difficulties in understanding the mother's mental illness caused them to feel surprised, shocked, helpless, fearful and frustrated and hindered them from effectively supporting mothers. "You felt bad. . .helpless. . .you couldn't do anything like. I couldn't help her", (SP5, father of two aged 39).

Some health visitors also recognised that partner's needed information about maternal mental illness, "they [partners] want help. . .support, they want you to tell them exactly what's

happening", (NHV1). The majority of partners also wanted health visitors to help them in supporting mothers, as they perceived that it was their responsibility to do so, "maybe she did need somebody. But then that should have been me. That shouldn't have been down to getting the health visitor to get somebody to come in. That should have been down to me to turn around and say well I'll be here" (SEP3, father of one aged 31). Partners also perceived that if they were provided with information about mental illness its overall effect on mothers would be reduced:

> I think probably a couple of days after the (partners name) was diagnosed properly with it, I would have liked somebody to sit us down and the first question I probably would have asked. . .what is postnatal depression? . . .and I think probably, if I had heard it from a professional, exactly what postnatal depression was I probably could have lessened that impact on (partner name) after that. . . (SP4, father of one aged 32).

### Theme 2: Health visitors' response to meeting families' needs

While health visitors described varying levels of engagement with families all three groups of participants indicated that FFP primarily entailed supporting mothers, while partners' needs were not addressed alongside this. Furthermore, while health visitors had mothers with both common and SMI on their caseloads they predominantly discussed a supportive role with mothers with PND rather than those with more complex conditions. The first subtheme focuses on two key activities that health visitors undertook, including supporting mothers via listening to them and supporting their babies mental and physical health. The second subtheme highlights the lack of support provided to partners by health visitors.

**Health visitor's response to mothers and their children.** One of the main supports that health visitors felt they could offer was discussing mother's concerns with them, "I would have done weekly listening visits, and we are curtailed to 5–6 weekly listening visits, I did many many more" (NHV10). Similarly, some mothers described how they felt supported by health visitors, reporting that they were "worth their weight in gold" (SM4, mother of one aged 30), or acted as "a crutch" (SEM3, mother of one aged 37). However, mothers also suggested their level of contact with health visitors varied with some reporting minimal contact while others had weekly or fortnightly contact, with some receiving additional daily phone calls.

In addition to supporting mothers, the majority of health visitors also perceived that they prioritised infant mental health as they recognised that some mothers may have difficulty in meeting their infants needs for communication and stimulation:

> Her interaction with her baby wasn't great. So she wasn't picking up on these babies cues and I suppose as a health visitor, we are big into infant mental health. And baby brain, and I suppose it's getting them to play and stimulate. And sometimes they're not doing that (SEHV4).

Health visitors supported babies directly by undertaking baby massage, and indirectly via supporting the mother, using the Solihull approach (i.e. a psycho-therapeutic approach, that focuses on the mother—infant relationship and assists parent's capacity to develop positive infant brain development) and by promoting skin-to-skin contact, breastfeeding and tummy time (i.e. time that baby spends on their stomach to strengthen muscles and motor skills).

**Health visitors' response to partners.** All health visitors acknowledged the significance of the partner in supporting the mother as reflected by SEHV7, "the daddy came out of work for a wee while there. . . because things were so bad. And he was great, it really helps the baby,

and then with that, it really helped mum". To facilitate the partner to support the mother some health visitors indicated that they helped them to understand mental illness. However, the majority of partners perceived that they were not supported in this way as reflected in the following quote:

> . . .she [mother] may not get an appointment [counsellor] say for another three weeks or a month. What happens between that time, who is the person who deals with [partner's name]. . .me, but there's no one talking to me about how I deal with it" (SP4, father of one aged 32).

The majority of participants across the three groups also perceived that few health visitors acknowledged partner's needs for support in their own right or actively supported them as reflected by a mother who stated, "It's just like ok right we'll sort the mother out and when she's ok right that's it who cares about the father" (WM1, mother of one aged 34).

## Theme 3: Health visitors' capacity to engage in FFP

Participants discussed the barriers that health visitor's experience in trying to address needs of mothers related to their mental illness and in supporting their partners. They also described how health visitors used their personal experience of PND and professional experience to primarily support mothers with this condition. The first subtheme identifies three key barriers for health visitors in providing optimal support to mothers and their families including: (1) health visitor's limited knowledge, skills and willingness to address needs of mothers related to their mental illness and to support their partners, (2) health visitors' large caseloads which reduce the amount of time they can spend with mothers and their families and (3) partners not being receptive to support provided by health visitors. The second subtheme outlines how health visitors used their personal and professional experience to engage and support mothers, albeit primarily those with PND.

**Health visitors' capacity to address mother's needs related to their mental illness and to support their partners.** Health visitors' perceptions of their capacity to engage in FFP was firstly influenced by their feelings towards supporting mothers with severe mental illness and their role around this. Most health visitors perceived that mental illness was a specialist area, and that mother's needs related to mental illness, other than PND, should be primarily addressed by mental health services as illustrated by the following quote, "Specifically when we are talking about postnatal depression, is up to the baby is. . .7–8 months. . .I think if it is after 7 months, its picked up then its maybe clinical depression, so then that's out of our remit" (SEHV4). Equally, some mothers also sensed the health visitor's uneasiness about discussing and addressing mental illness, indicating that they were "walking on egg shells", (WMI, mother of one aged 34), and "I think she knows enough to pick up the signs and be aware of what's going on. But that's about it really" (SM5, mother of two aged 35).

Health visitors also perceived that their workload was too high to allow them to effectively support mothers with mental illness including those with less severe forms, including PND:

> You're thinking oh gosh, em another one with postnatal depression, when am I going to slot her in because you cannot physically do three postnatal depressions in a day. Otherwise you are mentally drained. You sometimes look at your diary and think I really need to see you next week and I can't slot you in (SEHV4).

Similarly, another indicated, "I just think you have to have a minuscule caseload to be able to offer what that person needs as a whole" (SEHV2). Partners also indicated that a lack of

time and resources hindered health visitors from effectively supporting them also, "…I'm sure they are up to their eyes. They have enough people to see without trying to see husbands as well like. It's probably a time factor is it? Or budget factor…" (SP5, father of two aged 39).

The willingness and capacity of health visitors to engage partners also impacted the provision of FFP. While some health visitors engaged partners to obtain information about mothers, especially when there were child protection concerns, the majority were reluctant due to perceiving that it would breach the mother's confidentiality:

> You're going into the home, it's very difficult if the partner is there to start talking about mental health. … It's something …your always wary of and conscious of. That you don't want to be discussing that, because that's their own personal information (NHV8).

Additionally, mothers perceived that health visitors did not have the knowledge required to support partners, "It's not a selfish thing I just think there's no knowledge", (WMI, mother of one aged 34). However, both mothers and partners also suggested that in addition to partners being overlooked by health visitors, there was also a reluctance on their part to open up and accept support. For example, a father indicated, "…If your wife has postnatal depression, I don't think they really need to focus on the men like…" (SP4, father of one aged 32), and a mother stated, "men's mental health is not talked about as much as women" (SM5, mother of two aged 35). Fathers attributed their reluctance to seek support from health visitors for their own mental health to a widely held but erroneous perception in society that men do not suffer from mental illness and views on masculinity, "maybe you know I'm a man I don't suffer from them things. I don't know probably just the culture we're living in", (WP2, father of one aged 39).

**Health visitors draw upon professional and personal experience to support mothers.** Increasing professional experience enhanced health visitor's capacity to identify mother's problems and to be empathetic. Mothers described how health visitors drew upon their professional experience to identify when they were struggling. As one mother explained: "And it was her who knew straight away… I don't know if she is just that bloody experienced she can catch on to things or what" (SEM3, mother of one aged 37). In addition, mothers perceived that health visitor's professional experience allowed them to relate to and understand what they were going through:

> Because she is a professional in that industry em, and she has so many years' experience and she was able to relate to me. It was if sometimes she was able to read my mind. The things that she was saying. Eh, it was like oh my goodness how did you know I was thinking that or she would just put me at ease (SM4, mother of one aged 30).

In turn clinical experience helped health visitors to develop a positive relationship with mothers and to obtain their trust, which was seen as central to mothers being able to accept support, "I think the key thing in all my mums is to make the connection. If you get the connection and the rapport with these mums, they will seek you out and they will look for advice. And they will engage you" (SHV6). Some health visitors also described how they were able to use personal experience of PND in further supporting mothers and their families as it helped them to better understand their needs. Corroborating the importance of this personal experience, one mother indicated, "…I know she has experience of postnatal…I have got this in my head, that I need someone who knows what they're talking about" (WM1, mother of one aged 34).

## Discussion

This qualitative study contributes new knowledge about health visitors' FFP with mentally ill mothers and their families. Findings highlighted the restricted scope of health visitors' FFP. While health visitors provided care for mothers with various types and severity of mental illness, they predominantly addressed the mental health needs of mothers with PND; instead of also supporting mental health needs of mothers who had SMI. Furthermore, health visitors focused on supporting mothers and their children as opposed to also supporting partners. Supporting mothers' mental health took a toll on the health visitors themselves, with health visitors often feeling mentally drained. Health visitors provided this support in the context of excessive workloads and high caseloads of families. This may partly explain their restricted scope of practice with families impacted by maternal mental illness. Reid and Tracey [33] integrative review, exploring health visitors' workload, also highlighted how excessive caseloads can negatively impact the service they provide. For health visitors to increase their FFP they need to support mothers who have SMI more effectively and to consider how partners can be included in their practice in a manner that is both beneficial and acceptable to them. Equally, their excessive caseloads and limited knowledge and skills to support mothers with SMI needs further consideration.

### Conceptualization of the family and health visitors' engagement with fathers

The way health visitors conceptualised the family influenced their FFP. Health visitors perceived the family to refer to the mother-infant dyad and thus engaged primarily with them. While health visitors recognised that fathers could play a key role in supporting mothers, they did not perceive that they may require support themselves. While partners perceived themselves as providers and protectors of their family, and the best support for the mother, they felt misunderstood and perceived that their role in the family was not valued by health visitors. Previous research has reported health visitors' lack of engagement with fathers [4, 9, 18, 34, 35] and low rate of involvement of male partners in postnatal care services generally [36], despite policies such as the 'healthy child programme' [34] and [17] explicating the centrality of father's roles in child and family health. In the UK a study by Whitelock (2016) found that health visitors were quite open in labelling themselves as a mother focused service and did not believe that it was appropriate to support partners. Whitlock [35] concluded that at all levels of the health visiting service (practice and policy), it was deemed the 'norm' to work predominately with mothers and children, which reflected the historical culture within the service. Health visitors are primarily female and this has perpetuated the feminine nature of care provided within health visiting [37], and secondly perpetuates father's perceptions that the service is designed for mothers [18]. A global review of services has called for a more gender-balanced healthcare workforce within all professional categories [38]. Despite a lack of engagement between health visitors and partners, previous research suggests partners desire specifically tailored information to enable them to support the mother [39], our study also supports this assertion. Various approaches have been suggested to promote health visitor's engagement with fathers. These include, father-focused training [9, 40], changing health visitor's notions of fatherhood [34], engagement with fathers outside of the home visit including out-reach work [41], using digital platforms and utilising male friendly environments [38].

Fathers should be acknowledged as the providers and a much-needed support for the mother and family, if that is how they perceive themselves. At the same time, partners in the current research did not acknowledge that they required support to promote their own mental health despite some mothers perceiving that they did. Memon et al. [42] suggests that men

value self-reliance, particularly in relation to their health. Halle et al. [43] suggest that fathers' reluctance to accept support for their own mental health is exacerbated by expectations in society that fathers should protect their families. The current study provides further evidence that partner's perceptions that they do not need support for their own mental health is intertwined with issues of stigma, parenthood and masculinity.

## Stigma of mental illness and its impact on FFP

All three groups of participants and particularly mothers, discussed the stigma of mental illness. Reupert et al. [5] conducted a qualitative systematic review of family's experiences when a parent is hospitalised and found that mentally ill parents worry that their children will experience stigma within their community because of their illness. Ward et al. [16], in their interviews with professionals in adult mental health services, identified that stigma hindered involvement of family members. Namely, participants' perceived lack of family engagement was due to fear of stigma by association, thus they did not make further efforts to support family members. Research on mothers in mental health services suggests that many perceive they face discrimination and are viewed as unfit mothers, not listened to and undermined as a parent, resulting in reluctance to engage with services [44]. The current study supports these assertions about the impact of stigma on mother's engagement with professionals as well as furthering understanding of its impact on communication between mothers and their partners.

## Health visitor's use their lived and professional experience to support mothers who have PND

While health visitors experienced barriers to supporting mothers with SMI, both mothers and health visitors perceived that health visitors could use their own lived experience of PND to provide effective support. Lived experience of PND enabled health visitors to connect with mothers, have a better grasp of the issues they face and provide advice that was relatable. Lived experience is thought to assist professionals to empathise with service users, gain a deeper understanding, increase hope for recovery and reduce stigma [45]. Moreover, in the therapeutic arena, the concept of the 'wounded healer' [46] encourages the professional to recognise their own pain and vulnerability to enhance their skill [47]. However, there are some barriers to professional's using and disclosing personal experience of mental illness in practice. Although employers can no longer discriminate against those with mental illness [48], professionals may be cautious in disclosing mental illness due to queries around their capacity to practice effectively [49]. Moreover, shared experience does not necessarily lead to mutual understanding. For example, to assume that two individuals who have used mental health services will have shared experience and understanding is problematic, as it neglects the heterogeneity of both mental health and individual experience [50]. Therefore, health visitors must ensure their personal experience does not overshadow their professional ability to understand the individual experiences of the mother, thus recognising that all experiences are not the same. Also, while lived experience of PND can help health visitors support mothers with this condition, there is no evidence that this is transferrable to SMI or will lead to the promotion of FFP. Both mothers and health visitors also suggested that health visitor's increased years of professional experience enabled them to support mothers. There is limited evidence on the benefits of practice experience within the health visiting literature. However, health visiting has been likened to a progressive journey of change which requires time to adjust and adapt [51]. Thus, with increased time spent in practice, it may be expected that health visitors become more competent in their role. Increased professional experience has also been

associated with broader scope and extent of FFP in adult mental health services [6, 52]. Grant et al. [53] suggests that professional experience facilitated mental health nurses' FFP by enabling them to acquire effective interpersonal skills and knowledge which they used to form relationships with parents. These relationships then enabled the nurses to build trust with parents and support them in difficult circumstances, such as issues of child protection. There is also consensus in the literature that skills and competence are not enhanced solely through time spent in practice, but also through training and education [54].

## Limitations

Despite the novelty of this research in presenting multiple perspectives of health visitors' FFP, certain limitations should be noted. The majority of mothers within the sample had PND. Thus, it is not possible to say if there are any differences in FFP with mothers who had PND versus mothers with SMI. However, this was not the primary aim of the research and was also reflective of the wider population. There were also less partners recruited than initially antici- pated (approx. 10), however, the majority of codes had been generated by the fifth interview with partners, thus the requirements of data saturation were met [24]. Mothers and partners were heterosexual and primarily Caucasian, so results may not be applicable to non-heteronor- mative families, other types of non-traditional families and other races. The second author's gender may have influenced her interpretation of participants' experiences. As a female researcher, partners may not have been completely open with her in relation to their needs and experiences. It is possible, that the experiences they shared would have been different if they had been interviewed by a male researcher. Moreover, due to the perceived position of power as a researcher, it is possible that during the interviews the mothers were not fully open about challenges they faced out of fear of child protection procedures or fear of judgement.

## Conclusions

This study found that the scope of health visitors' FFP is restricted as they focus predominantly on mothers and children instead of also supporting partners. They are also unable to effectively support mother's and families' needs in relation to more SMI. The capacity of health visitors to adopt a whole of family approach is impacted by wider societal factors such as how the family is conceptualised, stigma surrounding mental illness, parental expectation and perceptions of roles in the family and health visitors' knowledge and skills to engage mothers who have SMI and partners. Currently these factors are not considered in policy, training or evaluation of health visitors' FFP despite current findings highlighting their importance.

The current study highlights that for health visitors to move beyond supporting mothers and their children to also supporting partners, how the family is conceptualised within health visiting services as a whole must be questioned. Health visitors also need to be supported to effectively engage partners through communication strategies that are appropriately tailored and targeted. A partner/father inclusive approach to national and local policy development could also help to guide health visitors in supporting them. Health visitors should also under- stand that for partners to more effectively support mothers, they need to share information about the mother's mental illness with them, after gaining mothers consent. Health visitors can gain mother's consent by forging a relationship to secure their trust and then explaining the benefits of involving partners in their care [6]. To more effectively support mothers with SMI, health visitors should also receive training and education to develop knowledge and skills in this area. Training should also encompass exploration of stigma, conceptualising the family and use of lived experience of mental illness in practice. The female centric nature of health visiting should also be addressed, to enable inclusivity of males. Finally, if health visitors are to

effectively support mothers who experience a range of mental illnesses and associated needs they must be allowed adequate time to do so. As the health visitor's caseloads, in the current study, exceeded the recommended limits of 250 children [55], there needs to be more effective recruitment and retention strategies within health visiting services in NI to expand the workforce.

## Acknowledgments

We thank the mothers and partners and professionals who completed the interview and the staff in the participating services for their assistance with recruitment and logistics.

## Author Contributions

**Conceptualization:** Anne Grant, Rachel Leonard, Mark Linden.

**Data curation:** Rachel Leonard.

**Formal analysis:** Anne Grant, Rachel Leonard.

**Investigation:** Anne Grant, Rachel Leonard, Mark Linden.

**Methodology:** Anne Grant, Rachel Leonard, Mark Linden.

**Project administration:** Rachel Leonard.

**Resources:** Rachel Leonard.

**Software:** Rachel Leonard.

**Supervision:** Anne Grant, Mark Linden.

**Validation:** Anne Grant, Rachel Leonard, Mark Linden.

**Visualization:** Anne Grant, Rachel Leonard.

**Writing – original draft:** Anne Grant.

**Writing – review & editing:** Anne Grant, Rachel Leonard, Mark Linden.

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
