## [Decision Letter · Decision Letter 0]

20 May 2024

PONE-D-24-06909A qualitative study of health visitors’ family focused practice with mothers with mental illness: perspectives of health visitors, mothers and spousesPLOS ONE

Dear Dr. Grant,

Thank you for submitting your manuscript to PLOS ONE. After careful consideration, we feel that it has merit but does not fully meet PLOS ONE’s publication criteria as it currently stands. Therefore, we invite you to submit a revised version of the manuscript that addresses the points raised during the review process.

We look forward to receiving your revised manuscript.

Kind regards,

Sudarshan Subedi

Academic Editor

PLOS ONE

Journal Requirements:

2. We note that you have referenced Bunting L, McCartan C, Davidson G, Grant A, McBride O, Mulholland C, et al. which has currently not yet been accepted for publication. Please remove this from your References and amend this to state in the body of your manuscript: (Bunting L, McCartan C, Davidson G, Grant A, McBride O, Mulholland C, et al. [Unpublished]) as detailed online in our guide for authors

Additional Editor Comments:

Thank you for submitting the manuscript. This is well written and I enjoyed reading this. In addition to the reviewer's comments, here are some comments.

- The title and/or the abstract should reflect where the study was carried out (country and region).

- Excellent introduction with relevant background information

- The design and method section indicates that this paper is based on the qualitative findings of a larger mixed methods study. However, it doesn’t say in details about the study approach of the qualitative part. Is it just a simple descriptive or analytical qualitative study or based on any theoretical approach?

- Sampling and data collection techniques are clear and specific. Better to write the average time (as stated 60 minutes) with minimum time and maximum time of the interview. I don't suggest using the term 'generally lasted'.

- Data analysis and rigour is well-explained.

- In the first para of 'discussion' section, it states "findings highlighted low level of FFP". This needs to be elaborated as the findings are qualitative and it's hard to say the low level in specific. It would be better if you provide some additional information in the data analysis section to support this claim. Since the low levels of FFP is one of the conclusions of the study, I suggest to make it clearer and more specific.

- I also suggest to illustrate the themes and relevant codes in a diagram as diagrammatic illustrations are very beneficial for the audience/readers to get a glimpse of the findings and guide them thoroughly in understanding the findings/discussion

Reviewers' comments:

Reviewer's Responses to Questions

**Comments to the Author**

1. Is the manuscript technically sound, and do the data support the conclusions?

Reviewer #1: Yes

2. Has the statistical analysis been performed appropriately and rigorously? 

Reviewer #1: N/A

3. Have the authors made all data underlying the findings in their manuscript fully available?

Reviewer #1: No

4. Is the manuscript presented in an intelligible fashion and written in standard English?

Reviewer #1: Yes

5. Review Comments to the Author

**Reviewer #1:** This is a very well-written manuscript covering an interesting and important topic. It gathers the perspectives of mothers, partners, and health visitors which provides a holistic view of the topic.

My main observation from reading this manuscript is that health visitors show dedication, expertise and compassion when treating mothers with mental illness (particularly PND). The main barrier to providing care for more serious mental illnesses and for treating partners would seem to stem from a shortage of resources (training, and particularly time). In the final sentence of the conclusion, it is mentioned that all the health visitors in the study exceeded the recommended workloads. I feel like this information should have been mentioned earlier as it provides important context for the findings.

I will now go through the manuscript with comments/suggestions on each section.

Title: Should it say ‘partners’ rather than spouses?

Abstract: I thought that the abstract was very comprehensive but could perhaps highlight the shortage of resources as mentioned above.

Introduction.

Page 4. A brief explanation of psycho education would be helpful.

Final sentence only mentions mothers and partners but the interview questions (on page 9) suggest that health visitors were also asked about the adequacy of support.

Design and methods.

I’m not sure whether the description of the survey stage of the study needs to be detailed here, but that is for the editors to advise. The authors might just need to refer to the survey results as providing criteria for selecting the sample of health visitors.

Page 6. The authors assigned pseudonyms to the participants but did not use them when reporting the findings. I wondered why this might be?

Data collection

Page 7, penultimate sentence says ‘both’ [perspectives] which implies 2 rather than the 3 that were gathered.

Page 8. Only the topic guide for the health visitor interviews is included. It would be helpful to see the topic guide(s) used in mothers’ and partners’ interviews.

Findings

Research participants. Page 12. It would be helpful to know how many of the health visitor participants were trained in FFP.

Themes

Pages 15 – 17. It would be easier to follow if some of the longer quotes were indented. There are also some long complex sentences which might be better broken up (e.g. the sentence beginning ‘Relatedly’ on page 16).

Page 18. The final sentence on the page is a good example of the dedication of health visitors.

Page 21. This section highlights resource issues relating to training (when is mental illness beyond their professional boundaries?) and workload. It was interesting that health visitors also alluded to their own needs (e.g. ‘mentally drained’) and I wondered if/how these were being met. This could be picked up in the discussion.

Page 23. This section highlights how health visitors are able to empathise with the women in their care. It made me wonder whether recommendations to employ more male health visitors were appropriate.

Discussion

P24. I realise I keep mentioning this, but I think that the issue of workload should be mentioned in the first paragraph. It might explain why health visitors prioritised the needs of mothers rather than partners’ needs.

P25. I liked the suggestions for promoting engagement with fathers. The findings indicate that partners would like information that would enable them to support mothers more effectively.

Fathers as financial providers is a bit of a gender stereotype – it needs to be clearer that that is their perception of themselves (if indeed that was the case for all partner participants).

Overall, I think that the manuscript is very well-presented, and I do not wish my suggestions to be taken as unduly critical. The authors are free to disregard my alternative interpretations of the findings which come from my own research with healthcare professionals who often struggle to provide the level of care that they would like to within time/resource constraints.

6. PLOS authors have the option to publish the peer review history of their article (what does this mean?). If published, this will include your full peer review and any attached files.

Reviewer #1: No

---

## [Author Response · Author response to Decision Letter 0]

7 Jun 2024

Dear sir/madam we have uploaded our response table.

---

## [Editor Report · Decision Letter 1]

26 Jun 2024

A qualitative study of health visitors’ family focused practice with mothers with mental illness in Northern Ireland: perspectives of health visitors, mothers and partners

PONE-D-24-06909R1

Dear Dr. Grant,

We’re pleased to inform you that your manuscript has been judged scientifically suitable for publication and will be formally accepted for publication once it meets all outstanding technical requirements.

Kind regards,

Sudarshan Subedi

Academic Editor

PLOS ONE

---

## [Editor Report · Acceptance letter]

6 Aug 2024

PONE-D-24-06909R1 

PLOS ONE

Dear Dr. Grant, 

I'm pleased to inform you that your manuscript has been deemed suitable for publication in PLOS ONE. Congratulations! Your manuscript is now being handed over to our production team.

Kind regards, 

on behalf of

Dr. Sudarshan Subedi 

Academic Editor

PLOS ONE